# Identification of Genes Associated with Sensitivity to Ultraviolet A (UVA) Irradiation by Transposon Mutagenesis of *Vibrio parahaemolyticus*

**Miki Maetani-Yasui** [1,†]**, Kazuaki Mawatari** [1,*,†] ⓘ**, Airi Honjo** [1]**, Thi Kim Ngan Bui** [1]**,**
**Takaaki Shimohata** [1]**, Takashi Uebanso** [1]**, Mutsumi Aihara** [2]**, Takahiro Emoto** [2]**,**
**Masatake Akutagawa** [2]**, Yohsuke Kinouchi** [2] **and Akira Takahashi** [1]

[1] Department of Preventive Environment and Nutrition, Institute of Biomedical Sciences, Tokushima University Graduate School, Kuramoto-cho 3-18-15, Tokushima City, Tokushima 770-8503, Japan; 3428hsm@gmail.com (M.M.-Y.); airi.102400@gmail.com (A.H.); kimnganvdd1190@gmail.com (T.K.N.B.); shimohata@tokushima-u.ac.jp (T.S.); uebanso@tokushima-u.ac.jp (T.U.); akiratak@tokushima-u.ac.jp (A.T.)

[2] Graduate School of Technology, Industrial and Social Sciences, Tokushima University, Minamijyousanjima-cho 2-1, Tokushima City, Tokushima 770-8506, Japan; aiharam@tokushima-u.ac.jp (M.A.); emoto@tokushima-u.ac.jp (T.E.); makutaga@tokushima-u.ac.jp (M.A.); kinouchi@tokushima-u.ac.jp (Y.K.)

[*] Correspondence: mawatari@tokushima-u.ac.jp; Tel.: +81-88-633-9598

[†] Miki Maetani-Yasui and Kazuaki Mawatari contributed equally to this work.

**Abstract:** Ultraviolet (UV) irradiation is used to disinfect water and food and can be classified as UVA (detected at wavelengths 320–400 nm), UVB (280–320 nm), and UVC (<280 nm). We developed a method for UVA sterilization of equipment with a UVA-light-emitting diode (LED); however, a high rate of fluence was needed to promote pathogen inactivation. The aim of this study was to identify genes associated with UVA sensitivity with the goal of improving UVA-LED-mediated bactericidal activity. We constructed a transposon-mutant library of *Vibrio parahaemolyticus* and selected six mutants with high sensitivity to UVA irradiation. Genes associated with this phenotype include F-type H$^+$-transporting ATPases (*atp*), as well as those involved in general secretion (*gsp*), and ubiquinone and terpenoid-quinone biosynthesis (*ubi*). Gene complementation resulted in decreased sensitivity to UVA-LED. The *atp* mutants had lower intracellular adenosine triphosphate (ATP) concentrations than the wild-type treatment, with 20 mM L-serine resulting in elevated ATP concentrations and decreased sensitivity to UVA-LED. The *gsp* mutants exhibited high levels of extracellular protein transport and the *ubi* mutants exhibited significantly different intracellular concentrations of ubiquinone-8. Taken together, our results suggest that the protein products of the *atp*, *gsp*, and *ubi* genes may regulate sensitivity to UVA irradiation.

**Keywords:** light-emitting diode; ultraviolet A; *Vibrio parahaemolyticus*

## 1. Introduction

*Vibrio parahaemolyticus* is a Gram-negative marine bacterial species that can cause acute gastroenteritis manifested by diarrhea, headache, vomiting, nausea, and abdominal pain; this is typically observed in individuals who have consumed raw or undercooked seafood [1–4]. *V. parahaemolyticus*-induced gastroenteritis is mediated by the type III secretion system and by enterotoxins, including thermostable direct hemolysin (TDH) and TDH-related hemolysin (TRH) [5–7]. Food poisoning associated with *V. parahaemolyticus* is a frequent occurrence worldwide [8,9].

Chlorination and ozonation have high efficiency against bacteria in general. However, some health problems have been observed. For instance, residual chlorine in drinking water can cause the formation of potentially carcinogenic halogenated by-products [10]. Likewise, ozonation can lead to the formation of high concentrations of undesired by-products, including bromates, which are also potential human carcinogens [11]. Other well-known disinfection methods include sunlight and ultraviolet (UV) irradiation; these modalities produce by-products, but those reported so far are below the level of health concerns [12]. UV rays are classified by wavelength into UVA (320–400 nm), UVB (280–320 nm), and UVC (<280 nm). Solar disinfection is an effective and inexpensive method of water treatment due to the fact that UVA and partial UVB in sunlight pass the ozonosphere and reach the surface of the Earth; this is not the case for UVC. UVC-based disinfection systems that utilize low-pressure mercury lamps are used widely as an effective sterilization method for both drinking and wastewater [13]. Because deoxyribonucleic acid (DNA) has a maximum absorption at approximately the same wavelength as UVC, irradiation can introduce photoproducts, including cyclo-butane pyrimidine dimers (CPDs) into the genomic DNA of target bacteria [14,15]. Some bacterial species have systems that repair thymine dimmers, including photolyases and the SOS response; these properties impart tolerance to the effects of UVC on bacterial DNA [16,17].

We originally developed a UVA irradiation disinfection system based on a light-emitting diode (UVA-LED) [14]. This system was capable of disinfection, and was specifically useful for the elimination of enteropathogenic bacteria, including *Escherichia coli* or *Vibrio parahaemolyticus* [18,19]. Our findings and those of other groups revealed that UVA irradiation induces cellular membrane damage and indirectly results in delayed growth by increasing intracellular levels of reactive oxygen species (ROS), including superoxide anion radicals ($O_2^{\bullet-}$), hydroxyl radicals ($OH^{\bullet}$), hydrogen peroxide ($H_2O_2$), and singlet oxygen ($^1O_2$) [14,18,19]. Furthermore, we recently reported that bacterial systems that were effective for repairing DNA damage secondary to UVC irradiation were not effective against damage elicited by UVA-LED irradiation [18]. Unfortunately, we found that UVA-LED irradiation had lower bactericidal efficiency than UVC did against *V. parahaemolyticus* [18,19]. As such, we performed the current study in order to learn more about bacterial sensitivity to UVA-LED. We were particularly interested in exploring factors underlying the UVA sensitivity of *V. parahaemolyticus*. As such, the aim of this study was to identify genes encoded by *V. parahaemolyticus* that are associated with UVA sensitivity. Toward this end, we constructed a *V. parahaemolyticus* mutant library using a transposon mutagenesis method; we then identified mutant strains with increased sensitivity to UVA-LED irradiation. In this study, we isolated and characterized six bacterial strains that displayed increased sensitivity to UVA-LED irradiation; among these were strains with mutations in genes encoding the $F_0F_1$-type ATP synthase subunits (*atp*) as well as those involved in the biosynthesis of ubiquinone (*ubi*) and the general secretion pathway (*gsp*) genes. The goal of this study was to clarify how to modulate sensitivity to UVA-LED irradiation by the genes and gene products.

## 2. Materials and Methods

### 2.1. Microbial Strains and Cell Preparation

The wild-type (WT) strain of *V. parahaemolyticus*, RIMD2210633, was obtained from the Research Institute for Microbial Diseases (RIMD), Osaka University, Japan [20]. The bacteria and plasmids used in this study are listed in Table 1. Genetically modified organisms (GMOs) are regulated under the Cartagena Domestic Law. This study was conducted in conformity with the guidelines for the care and use of GMOs of the Institute of Biomedical Sciences, Tokushima University Graduate School (No. 27-375). Bacteria were cultured in Luria-Bertani (LB) broth or on LB agar plates, which contain 1% tryptone (BD, Franklin Lakes, NJ, USA), 0.5% yeast extract (BD), and 0.5 to 3% NaCl (Nacalai Tesque, Kyoto, Japan). WT and mutant strains were cultured at 37 °C for 18 h in order to reach stationary phase; antibiotics were added as appropriate.

**Table 1.** Bacterial strains and plasmids used in this study.

| Strain and Plasmid | Description | Source or Reference |
|---|---|---|
| | *Vibrio parahaemolyticus* strains | |
| RIMD2210633 | KP-positive, serotype O3:K6; clinical isolate; Wild-type strain in this study | Makino et al. [20] |
| *VP0097* | RIMD2210633 mutant #0002, Ez-Tn5 DHFR-1 insertion to *VP0097*, Tmp^r | This study |
| *VP0136* | RIMD2210633 mutant #0358, Ez-Tn5 DHFR-1 insertion to *VP0136*, Tmp^r | This study |
| *VP0140* | RIMD2210633 mutant #0521, Ez-Tn5 DHFR-1 insertion to *VP0140*, Tmp^r | This study |
| *VP0315* | RIMD2210633 mutant #0092, Ez-Tn5 DHFR-1 insertion to *VP0315*, Tmp^r | This study |
| *VP3069* | RIMD2210633 mutant #0078, Ez-Tn5 DHFR-1 insertion to *VP3069*, Tmp^r | This study |
| *VP3075* | RIMD2210633 mutant #0229, Ez-Tn5 DHFR-1 insertion to *VP3075*, Tmp^r | This study |
| | Plasmids | |
| pSA19CP | Expression vector; Cm^r | Nakano et al. [21] |
| p*VP0097* | pSA19CP expressing *VP0097*, controlled by the *tdhA* promoter; Cm^r | This study |
| p*VP0136* | pSA19CP expressing *VP0136*, controlled by the *tdhA* promoter; Cm^r | This study |
| p*VP0140* | pSA19CP expressing *VP0140*, controlled by the *tdhA* promoter; Cm^r | This study |
| p*VP0315* | pSA19CP expressing *VP0315*, controlled by the *tdhA* promoter; Cm^r | This study |
| p*VP3069* | pSA19CP expressing *VP3069*, controlled by the *tdhA* promoter; Cm^r | This study |

Note: KP, Kawasaki phenomenon; Cm^r, chloramphenicol resistant; Tmp^r, trimethoprim resistant; *tdhA*, thermostable direct hemolysin A; DHFR-1, dihydrofolate reductase-1.

## 2.2. Construction of the V. parahaemolyticus Mutant Library and Screening of the Mutants for High Sensitivity to UVA

A transposon mutation kit, Ez-Tn5™ DHFR-1 Tnp Transposome kit (Epicentre Biotechnologies, Madison, WI, USA), was used in this study. After an 18-h culture in LB broth at 37 °C, the WT strain was diluted 20 times in LB broth containing 0.5% NaCl and cultured at 37 °C for an additional 2.5 h. The cells were collected by centrifugation (7740× $g$ for 2 min) and washed three times with 1 mL of ice-cold electroporation (EP) buffer, containing 272 mM sucrose, 1.12 mM $NaH_2PO_4$, 5.88 mM $NaHPO_4$, 1 mM MgCl2, and 7% dimethyl sulfoxide (DMSO). The cells were then suspended in 100 μL of EP buffer containing 1% TransposomeTM solution and 1% TypeOne™ Restriction Inhibitor (Epicentre Biotechnologies) at 4 °C for 10 min and transformed by EP using a Gene Pulse apparatus (Bio-Rad Laboratories, Hercules, CA, USA), at 1000 Ohms, 25 μF, and 1300 V. Following EP, the cells were transferred into 500 μL of LB broth containing 1% NaCl and incubated for 1.5 h at 37 °C. To isolate the Ez-Tn5™ DHFR-1 transposon mutants, the cells were incubated on LB agar containing 1% NaCl and 15 μg/mL trimethoprim (TMP) at 37 °C for 18 h. The isolated colonies were cultured individually with LB containing 3% NaCl and 15 μg/mL TMP and stored frozen in 8% glycerol. Mutants with high sensitivity to UVA were identified using a two-step screening process, as indicated in Figure 1. To identify the mutation-inserted genes, genomic DNA of the high UVA-sensitive mutants was purified using a QIAamp DNA mini kit (Qiagen, Valencia, CA, USA); the DNA sequences flanking the Ez-Tn5™ DHFR-1 transposon were evaluated according to the manufacturer's instructions.

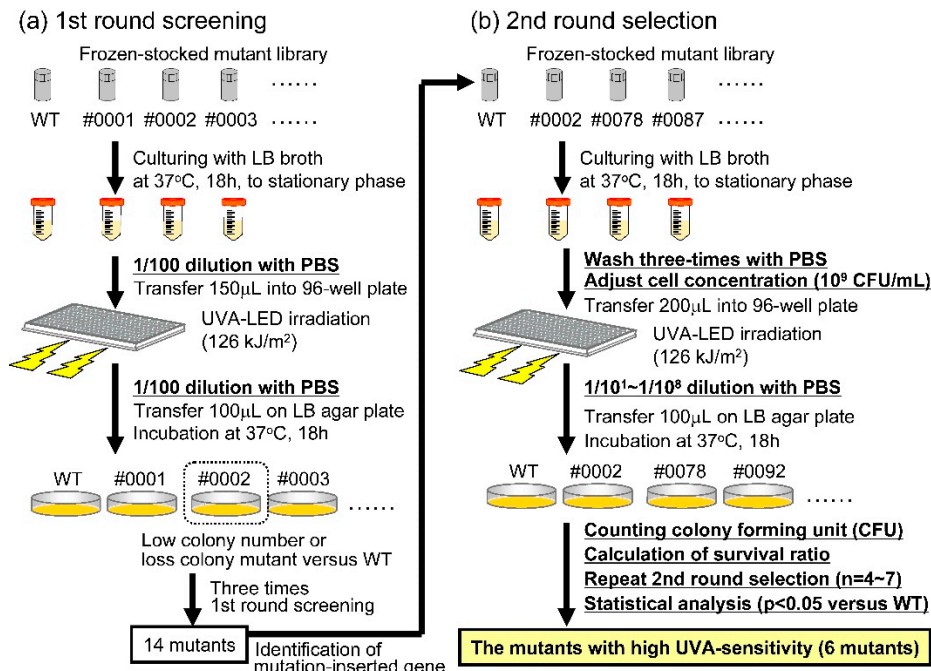

**Figure 1.** Selection of highly UVA-sensitive mutants from a *V. parahaemolyticus* transposon-mutant library. Highly UVA-sensitive mutants were separated from the mutant library by a two-step screening process. (**a**) First-round screening as shown. The bacterial cell numbers in each of the samples were not adjusted prior to UVA irradiation and the number of colony-forming units (CFUs) detected after culture of the UVA-irradiated samples was not determined. (**b**) Second-round screening. The bacterial cell concentrations were adjusted to $10^9$ CFU/mL prior to UVA irradiation; CFUs in each sample both with or without UVA irradiation were counted.

## 2.3. Gene Complementation in the Mutants

Relevant genes were amplified by polymerase chain reaction using the primers listed in Table 2 and cloned into the pSA19CP vector [19,21]. Plasmids were introduced into the transposon mutants via EP using a Gene Pulse apparatus (Bio-Rad); transfected bacteria were selected on LB agar containing 10 μg/mL chloramphenicol (Cm). Gene expression in these plasmids was under the control of the thermostable direct hemolysin A (*tdhA*) gene promoter; this promoter is well-characterized and is highly active during stationary growth [7,21].

**Table 2.** Oligonucleotide primers for gene complementation.

| Gene | Sequence (5′–3′) | |
| :---: | :---: | :---: |
| | **Forward** | **Reverse** |
| *VP0097* | 5′–**TCTAGA** ATGACGCCAGCAGAATTAAAGC–3′ | 5′–**GAATTC** CTATTGACGATAAGCTCGCCAAC–3′ |
| *VP0136* | 5′–**TCTAGA** ATGAAATTTAAGCGTAGTAAGC–3′ | 5′–**GAATTC** TTATTGGAAGTCTTGCATGTTCCA–3′ |
| *VP0140* | 5′–**TCTAGA** ATGGCTAATCGTCAGCGCGGT–3′ | 5′–**GAATTC** CTACTCAGCCGAACGGTCAGAAA–3′ |
| *VP0315* | 5′–**TCTAGA** ATGCACAACAAAATACAACCC–3′ | 5′–**GGATCC** TCACGAGCGCTGGTCATA–3′ |
| *VP3069* | 5′–**GGATCC** ATGGCTACAGGTAAGATCGTAC–3′ | 5′–**GAATTC** TTATAGCTTCTTCGCATTCTCG–3′ |
| *VP3075* | 5′–**TCTAGA** ATGGCTGCGCCAGGTGAA–3′ | 5′–**GAATTC** TTAATGATCAGAGTCTTCGTGTGC–3′ |

Note: Restriction sites are in bold type.

### 2.4. UV Irradiation

UVA light was emitted by a UVA-LED device developed as previously reported [18,19]. UVA-LED was pointed upward from the bottom of a 96-well plate; light intensity was maintained at 420 W/m$^2$ at the bottom of the well with an average peak wavelength of 365 nm (Supplementary Material Figure S1a). UVB or UVC light was emitted downward and pointed at the surface of the solution in each well using a low-pressure UV lamp (3UV-38; UVP, Upland, CA, USA) at an intensity of 0.9 or 0.7 W/m$^2$ and with average peak wavelengths of 302 and 254 nm, respectively (Supplementary Material Figure S1b,c). UV intensities were measured using a multiple wavelength photometer (MCPD 3700A; Otsuka Electronics, Osaka, Japan) and the handheld power meter NOVA II (Ophir Japan, Tokyo, Japan).

### 2.5. Measurement of Sensitivities to UV Irradiation

Sensitivities to UV irradiation were determined as bactericidal activity using a colony-forming assay. *V. parahaemolyticus* strains were collected by centrifugation (7740× *g*, 2 min, 25 °C) and washed three times with sterile phosphate-buffered saline (PBS). Then, 15 min before UV irradiation, the bacterial cells were treated with or without 20 mM L-serine (Supplementary Material Figure S2). The WT or the mutant strains were both adjusted to 10$^9$ CFU/mL in PBS. The irradiant fluences of UVA, UVB, and UVC were 126, 0.27, and 0.063 kJ/m$^2$, respectively. After UV irradiation, bacterial suspensions were diluted appropriately, plated on LB agar plates with or without 20 mM L-serine, and incubated at 37 °C for 18 h. Following the incubation, the colonies were counted, and the log of the survival ratio was calculated as follows:

$$\text{Log survival ratio} = \log_{10}(N_t/N_0), \tag{1}$$

where $N_t$ is the colony count of the UV-irradiated sample, and $N_0$ is the colony count of the sample that had not undergone UV irradiation.

### 2.6. Intracellular ATP Concentration

Intracellular concentrations of ATP were measured for each of the *V. parahaemolyticus* mutants using a Kinsiro ATP Luminescence kit (TOYO B-Net, Tokyo, Japan) in accordance with the manufacturer's instructions. After exposure to 20 mM L-serine or the diluent control, the mutant bacteria were collected by centrifugation (7740× *g*, 2 min, 25 °C), washed twice with PBS, resuspended in 1.2 mL of PBS, and stored at −80 °C until measurements could be performed. To measure intracellular ATP concentrations, samples were homogenized by ultrasound (Branson Ultrasonic, Danbury, CT, USA). After centrifugation (22,000× *g*, 10 min, 4 °C), the supernatants remaining were normalized for protein concentration using the bicinchoninate (BCA) method (Pierce, Rockford, IL, USA). Then, 20 μL of each sample that was normalized for total protein concentration was mixed with 180 μL luciferase/luciferin reaction solution; luminescence intensities were measured using a Varioskan Flash (Thermo Fisher Scientific, Waltham, MA, USA).

### 2.7. Sodium Dodecyl Sulfate-Polyacrylamide Gel Electrophoresis (SDS-PAGE) and Silver Staining

After bacteria reached the stationary phase (grown for 18 h in LB broth at 37 °C), the cultures were divided into pellets and supernatants by centrifugation (7740× *g*, 2 min, 25 °C) to facilitate the detection of both intracellular and extracellular protein, respectively. In order to remove any residual intact bacterial cells, the supernatants were filtered with 0.22-μm pore filters and mixed with 10% trichloro acetate, which resulted in the 10-fold concentration of extracellular protein. After centrifuging at 17,500× *g* for 10 min at 4 °C, pellets were washed twice with ice-cold acetone. The extracted extracellular or intracellular proteins were separated using SDS-PAGE and visualized by silver staining (ATTO, Tokyo, Japan).

### 2.8. Ubiquinone-8 Measurements

For quantitative determination of UQ-8, the lipid phase was extracted from WT and mutant strains and analyzed liquid chromatography-time of flight mass spectrometry (LC-TOFMS). After culturing to the stationary phase in LB broth for 18 h at 37 °C, WT or the mutants were collected by the centrifugation ($7740 \times g$, 2 min, 25 °C), and washed three times with sterile PBS; the lipid phase was then extracted as previously reported [22]. The samples were analyzed by an Agilent LC-TOFMS system 6200 with ZORBAX Eclipse Plus C18 column (Agilent Technologies, Santa Clara, CA, USA) at a flow rate of 1 mL/min using water-methanol (40:60) as the initial mobile phase. After sample injection, the percentage of methanol was increased to 100% over 10 to 30 min and then maintained for an additional 20 min. Agilent jet-stream electrospray ionization (Dual AJS ESI) was used as a source of ions. The ESI source was operated in positive mode, including spray voltages of 3500 V for the capillary entrance and 500 V for the nozzle, with a nitrogen sheath gas temperature at 250 °C and a flow rate of 12 L/min. Nitrogen drying gas was introduced at 150 °C at a flow rate of 10 L/min, with the nitrogen nebulizer at 45 psig. UQ-8 purified from *E. coli* (Avanti Polar Lipids, Alabaster, AL, USA) was used to generate a standard curve for determining UQ-8 concentrations. An extracted ion chromatogram (EIC, ± 1 ppm) of UQ-8 and integral analysis of peak areas were generated by Agilent MassHunter software (Agilent Technologies).

### 2.9. Statistical Analysis

Statistical analyses included ANOVA with Bonferroni's multiple comparison tests using Statview 5.0 software (SAS Institute Inc., Cary, NC, USA). Student's t-tests were used for paired data when appropriate. Values of $p < 0.01$ or $p < 0.05$ were considered to be statistically significant.

## 3. Results

### 3.1. Screening of the Mutants with High UVA Sensitivity

Highly UVA-sensitive mutants were identified in the mutant library using a two-step screening process (Figure 1). Fourteen mutants were selected as candidates for high sensitivity to UVA irradiation after the first round of screening. In the second round of screening, eight mutants were identified that exhibited similar or lower sensitivities to UVA irradiation that was exhibited by the WT (Supplementary Material Figure S3). The other six mutants were identified as highly sensitive to UVA irradiation; these mutants include those whose chromosomal DNA was modified by transposon insertion into VP0097 (#0002), VP0136 (#0521), VP0140 (#0358), VP0315 (#0092), VP3069 (#0078), or VP3075 (#0229; Figures 2 and 3a). Information provided by the National Center for Biotechnology Information, Kyoto Encyclopedia of Genes and Genomes, and RIMD facilitated the classification of these six mutant strains into three functional/orthologous groups. Specifically, VP3069 and VP3075 were functionally classified as being associated with $F_0F_1$-type ATP synthase, VP0136 and VP0140 with general protein secretion pathways, and VP0097 and VP0315 with the biosynthesis of ubiquinone and/or other terpenoid-quinones (Table 3).

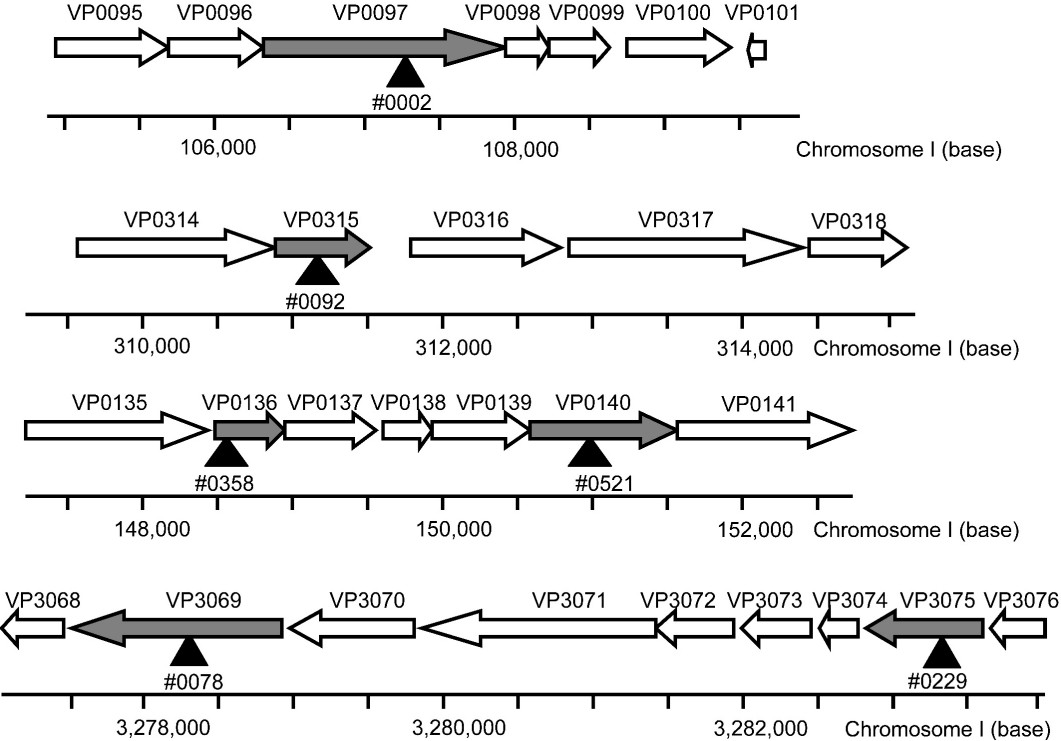

**Figure 2.** Map of *Vibrio parahaemolyticus* genes associated with the highly UVA-sensitive mutants. Arrows indicate open reading frames of *V. parahaemolyticus* genes. Arrow heads and gray arrows indicate mutant genes associated with high UVA sensitivity; the white arrows denote other genes with Ez-Tn5 DHFR-1-insertions.

**Table 3.** Putative product and functional classification of the Ez-Tn5 DHFR-1-inserted genes in highly UVA-sensitive mutants.

| Genes | Orthologous Genes | Product | Functional Classification |
|:---:|:---:|:---:|:---:|
| VP3075 | atpB | ATP synthase $F_0F_1$ subunit alpha | Oxidative phosphorylation |
| VP3069 | atpD | ATP synthase $F_0F_1$ subunit beta | F-type $H^+$-transporting ATPase |
| VP0136 | gspG | general secretion pathway protein G | Bacterial secretion system |
| VP0140 | gspK | general secretion pathway protein K | Type II secretion system |
| VP0097 | ubiB | ubiquinone biosynthesis protein UbiB | Ubiquinone and other |
| VP0315 | ubiX | 3-polyprenyl-4-hydroxybenzoate carboxy-lyase UbiX | Terpenoid-quinone biosynthesis |

Note: ATP, adenosine triphosphate.

## 3.2. Sensitivities to Different UV Wavelengths

To investigate relative sensitivities to different wavelengths of UV irradiation, the six mutants selected were irradiated with UVA, UVB, or UVC, and the survival ratios were measured. All of the six mutants were more sensitive to the lethal impact of UVA irradiation than the WT; their sensitivities to UVB and UVC could not be distinguished from those of the WT (Figure 3). From these results, we conclude that the UV sensitivities of the mutants *VP0097*, *VP3069*, *VP0136*, *VP0315*, *VP3075*, and *VP0140* might be restricted to UVA-associated wavelengths.

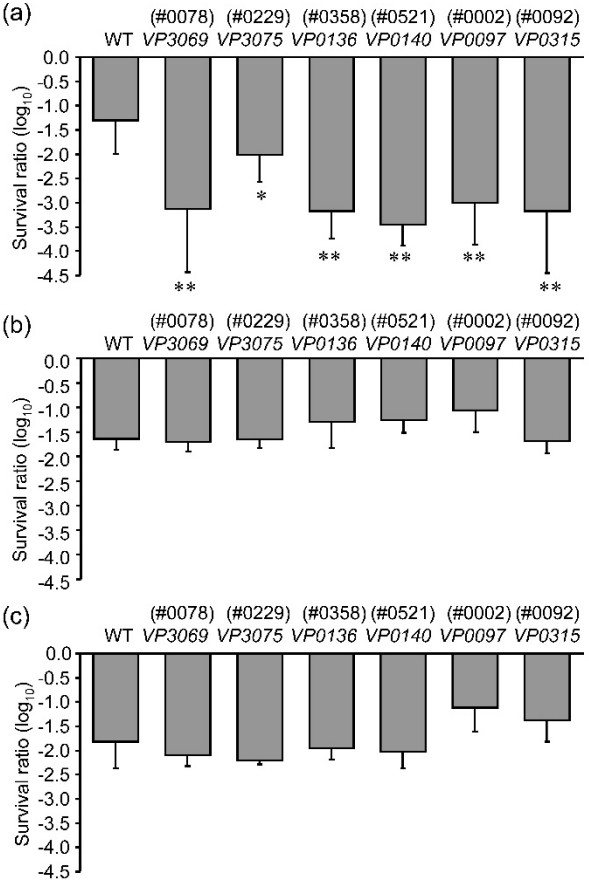

**Figure 3.** Sensitivities of six *V. parahaemolyticus* mutants to different wavelengths of UV light. (**a**) Responses to UVA-LED irradiation at 126 kJ/m$^2$, (**b**) responses to UVB at 0.27 kJ/m$^2$, (**c**) responses to UVC at 0.063 kJ/m$^2$. The sensitivities to UV irradiation were presented as a logarithmic (log$_{10}$) analysis of the survival ratio as described in the materials and methods. The detailed emission spectra of UVA-LED, UVB, and UVC are shown in Supplementary Material Figure S1. Values shown are means ± SD; n = 3–6, where n = number of independent replicates. * $p < 0.05$ or ** $p < 0.01$ versus WT.

*3.3. Evaluation of Intracellular ATP Concentration and Sensitivity to UVA Irradiation in Strains with Mutations in $F_0F_1$-Type ATP Synthase Genes*

To investigate the relationship between sensitivity to UVA and mutations in genes encoding $F_0F_1$-type ATP synthase, we measured the intracellular ATP concentrations in the mutants *VP3069* and *VP3075*. Intracellular ATP concentrations were significantly lower in both *VP3069* and *VP3075* mutants than in the WT (Figure 4a,b). The gene complementation of *VP3069* (p*VP3069/VP3069*) or *VP3075* (p*VP3075/VP3075*) restored the intracellular ATP concentrations and decreased UVA sensitivity, although not to the same extent as observed in the WT (Figure 4a). To determine the relationship between the intracellular ATP concentration and UVA sensitivity, both mutant strains were treated with L-serine (Supplementary Material Figure S2) [23]. The 20 mM L-serine treatment increased the intracellular ATP concentrations and decreased the UVA sensitivities in both *VP3069* and *VP3075* (Figure 4b,d). These results suggested that intracellular ATP concentrations regulated by $F_0F_1$-type ATP synthase might be associated with sensitivity to UVA.

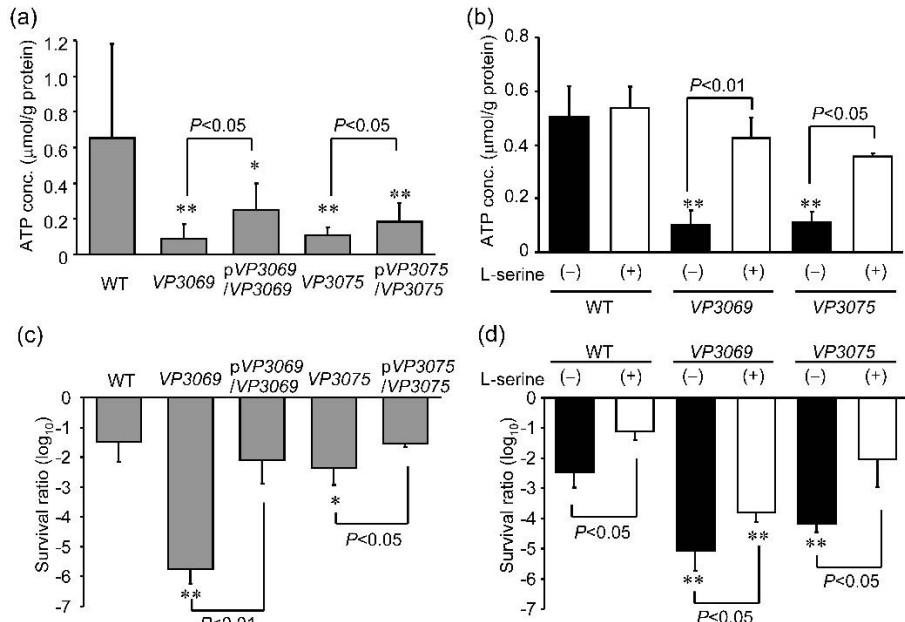

**Figure 4.** Intracellular ATP concentration and sensitivity to UVA irradiation in the mutants of $F_0F_1$-type ATP synthase genes. (**a,b**) ATP concentrations in response to gene complementation of $F_0F_1$-type ATP synthase gene mutations, p*VP3069/VP3069* and p*VP3075/VP3075*. (**c,d**) WT and mutant strains in the absence (■) or presence of 20 mM L-serine (□). Intracellular ATP concentration was measured by a luciferase/luciferin reaction as described in the materials and methods section. (**c,d**) Sensitivity to UVA irradiation was measured as described in the materials and methods section. Values shown are means ± SD (n = 3–7, n = number of independent experimental replicates). * $p < 0.05$ or ** $p < 0.01$ versus WT.

### 3.4. Evaluation of Extracellular Protein Content and Sensitivity to UVA among Mutants of General Secretion Pathway Genes

To investigate the relationship between sensitivity to UVA and the mutation of the general secretion pathway gene, we measured the extracellular protein of the *VP0136* and *VP0140* mutants. The content of extracellular protein in the *VP0136* and *VP0140* mutants was significantly higher than that in WT (Figure 5a), but the intracellular proteins were not different among WT and the mutants (Figure 5b). The gene complementation of VP0136 (p*VP0136/VP0136*) or VP0140 (p*VP0140/VP0140*) significantly decreased the content of extracellular protein and increased UVA sensitivity (Figure 5a,c), whose levels were the same levels in WT (Figure 5a,c). From these results, membrane permeability controlled by the proteins of general secretion pathway might be associated with UVA sensitivity.

### 3.5. Intracellular Concentrations of Ubiquinone-8 and Sensitivity to UVA Irradiation among Strains with Mutations in Ubiquinone Biosynthesis Genes

In *E. coli* and *Vibrio* spp., the UQ tail includes eight isoprene groups and as such is designated UQ-8 [24]. Biosynthesis of UQ-8 in *E. coli* relies on *ubi* genes [25]. In order to explore the relationship between sensitivity to UVA and mutations in genes responsible for ubiquinone biosynthesis, we measured intracellular concentrations of UQ-8 ($C_{51}H_{72}O_2$) in *VP0097* and *VP0315* by LC-TOFMS. The positive ions associated with UQ-8 (mass-to-charge ratio [m/z] = 727.5673 [M + H] and 749.5490 [M + Na]) were detected at a retention time = 44.8 min (Figure 6a). We detected a lower concentration of UQ-8 in the *VP0315* mutant than in the WT; interestingly, the *VP0097* mutant exhibited a higher UQ-8 concentration than the WT (Figure 6a,b). The gene-complemented strains, p*VP0097/VP0097* and p*VP0315/VP0315*, maintained levels of both the intracellular UQ-8 concentration and UVA sensitivity that were similar to the WT (Figure 6b,c). From these results, we conclude that both VP0097 and VP0315 serve to regulate UQ-8 concentrations and also the sensitivity to UVA irradiation.

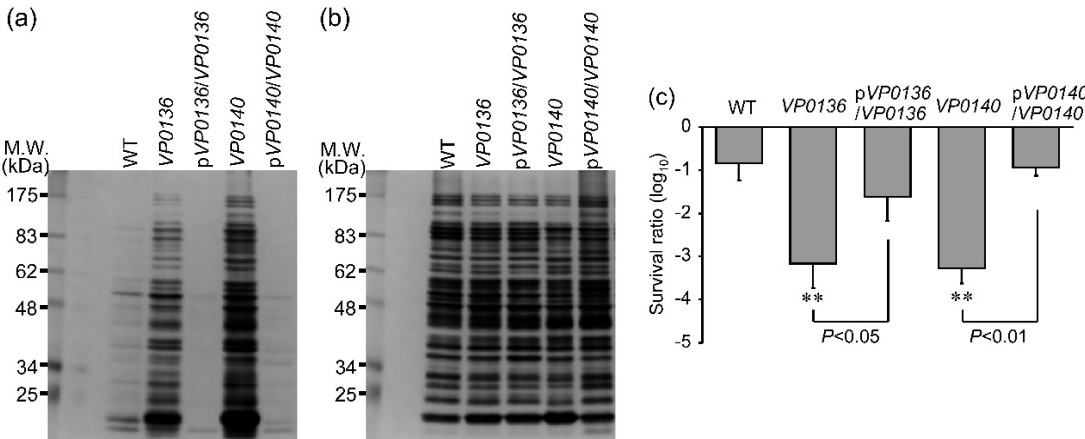

**Figure 5.** Extracellular and intracellular protein content and sensitivity to UVA irradiation among strains with mutations in genes association with general secretion pathways. (**a,b**) A typical image of protein content revealed by silver staining after SDS-PAGE. (**a**) Extracellular proteins and (**b**) intracellular proteins were separated as described in the materials and methods section. (**c**) Sensitivity to UVA irradiation, as shown in Figures 2a and 3c. Sensitivities of gene-complemented strains of general secretion pathway genes, p*VP0136/VP0136* and p*VP0140*//, were also evaluated. Values shown are means ± SD (n = 3–5, n = number of independent experimental replicates; ** $p < 0.01$ versus WT.

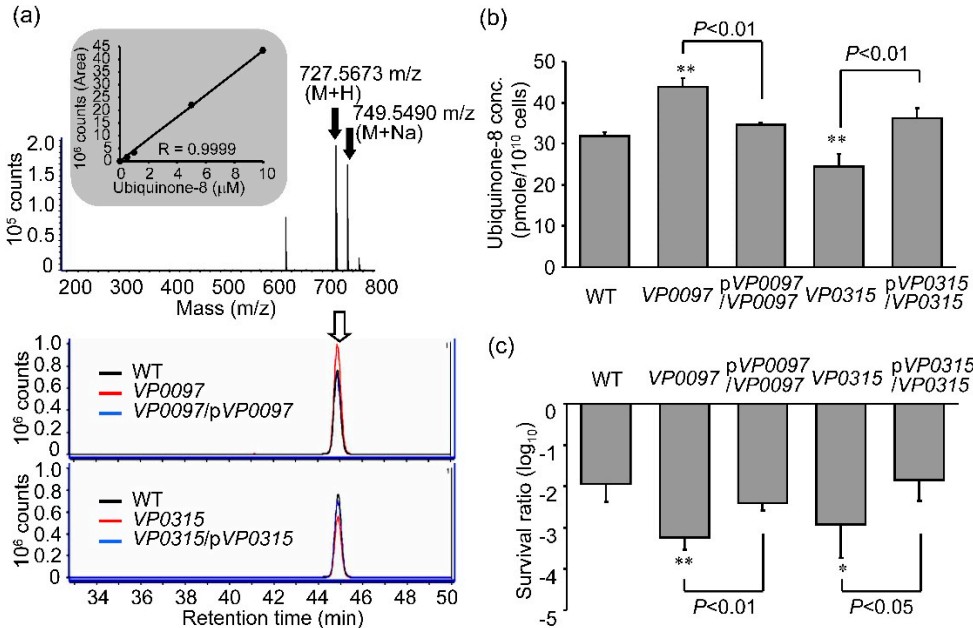

**Figure 6.** Intracellular ubiquinone-8 concentration and sensitivity to UVA irradiation associated with strains with mutations in ubiquinone biosynthesis genes. (**a**) Mass spectra in the target peak area (white arrow, retention time = 44.8 min) and extracted ion chromatograms (EICs) of ubiquinone-8 (UQ-8, $C_{51}H_{72}O_2$, m/z = 727.5665 [M + H]). Gray panel documents the standard curve featuring UQ-8 purified from *Escherichia. Coli.* (**b**) Intracellular concentrations of ubiquinone-8. (**c**) Sensitivity to UVA irradiation. Intracellular UQ-8 was measured by liquid chromatography-time of flight mass spectrometry (LC-TOFMS) as described in the materials and methods section. Values shown are means ± SD (n = 3–5, n = number of independent replicates). * $p < 0.05$ or ** $p < 0.01$ versus WT.

## 4. Discussion

To identify bacterial genes associated with UVA sensitivity, we constructed a transposon-mutant library and selected mutant strains of *V. parahaemolyticus* that exhibited high sensitivity to UVA

irradiation. We selected six mutants that were highly sensitivity to UVA irradiation only, and did not exhibit increased sensitivity to either UVB or UVC. The six selected mutant strains included the mutations in genes encoding F-type $H^+$-transporting ATPase (#78; *VP3069*, #229; *VP3075*), targets in the general secretion pathway (#521; *VP0136*, #358; *VP0140*), and those involved in ubiquinone and other terpenoid-quinone biosynthesis (#2; *VP0097*, #92; *VP0315*). Gene complementation studies performed with these mutant strains resulted in decreased sensitivity to UVA irradiation, approaching levels comparable to the WT. These results suggest that the products of these aforementioned genes serve to regulate sensitivity to UVA irradiation. We previously reported that UVA-LED irradiation increased levels of ROS in *E. coli.* and *V. parahaemolyticus* [14,19]. In addition, all of the six UVA-sensitive mutants had highly sensitivity to hydrogen peroxide (Supplementary Material Figure S4). These suggest that the genes are associated with the sensitivity to both UVA irradiation oxidative stresses.

The enzyme ATP synthase generates ATP from ADP and phosphoric acid via the rotation of the $F_0$ motor on biological membranes with a hydrogen ion concentration gradient [26]. We detected remarkable decreases in intracellular ATP concentrations in the mutant strains *VP3069* and *VP3075* that involve the $F_0F_1$-type ATP synthase gene; ATP concentrations increased in response to gene complementation (Figure 4a). Although the resistance to UVA irradiation was restored by complementation, the intracellular ATP concentration in the mutants remained lower than that observed in WT cells. Franziska et al. [27] reported that intracellular ATP concentrations decreased in response to UVA irradiation and that the concentration was related to resistance against UVA in *E. coli.* Furthermore, Bosshard et al. [28] reported that the alpha-subunit of the $F_0F_1$-type ATP synthase underwent aggregation in response to UVA irradiation. From these results, we conclude that $F_0F_1$-type ATP synthase may play an important role in supporting the intracellular ATP concentration during periods of UVA irradiation/stress. However, the L-serine treatment increased the intracellular ATP concentration and decreased the sensitivity to UVA irradiation (Figure 4b,d). L-serine treatment increases the ATP concentration through one-carbon metabolism, which is independent of $F_0F_1$-type ATP synthase [23]. Akhova et al. [29] reported that the ATP/ADP ratio was changed by oxidative stress under antibiotic treatment in *E. coli.* From these results, the intracellular ATP concentration controlled by not only $F_0F_1$-type ATP synthase but also other pathways, including one-carbon metabolism, may be crucial factors for modulating UVA sensitivity.

General secretion is also known as the Sec-dependent or the type II secretion system (T2SS). Many Gram-negative bacteria utilize the T2SS to facilitate translocation of folded proteins from the periplasm through the outer membrane and likewise inwards from the extracellular environment [30]. Expression of outer membrane proteins (Omps), OmpU, OmpT, and OmpS, were all decreased and the outer membrane integrity was compromised in T2SS mutants of *V. cholerae* [31,32]. In this study, we observed a drastic increase in the extracellular content associated with mutations in general secretion pathway genes in *V. parahaemolyticus* (i.e., *VP0136* and *VP0140*; Figure 5a); Sikora et al. [33] reported similar findings in *V. cholerae*. In addition, this group determined that *V. cholerae* T2SS mutants were more sensitive to hydrogen peroxide and endogenous ROS formation than the WT strain [33]. From these results, factors associated with T2SS-regulated outer membrane integrity are likely to be very important for sensitivity to UVA and oxidative stress. Apart from interactions within the outer membrane and T2SS proteins, these latter components may also interact with other constituents of the cell envelope. As but one example, alterations in the structure of lipopolysaccharide (LPS) have an immediate impact on the function of the T2SS system in *Pseudomonas aeruginosa* [34,35]. Recently, Johnson et al. [36] reported that T2SS from *V. cholerae* supported biofilm formation. Biofilms are microbial communities that are embedded in polysaccharides, nucleic acids, and associated extracellular protein matrix; they function to protect the resident bacteria from predators as well as from the effects of extracellular stress, the actions of antibiotics, and clearance by the immune system [37–39]. Pezzoni et al. [40] reported that biofilms containing *P. aerginosa* were more resistant to UVA irradiation than planktonic cells in culture. From these results, we conclude that biofilms controlled by the general secretion pathway of *V. parahemolyticus* may also be associated with sensitivity to UVA irradiation.

Ubiquinone (UQ), also known as coenzyme Q, is a lipophilic metabolite identified in organisms ranging from bacteria to mammals that includes a conserved quinone head group and an isoprenoid hydrophobic tail that varies in length among different species [41]. UQ-8 is located in the bacterial plasma membrane and has been described as an essential element underlying aerobic respiratory growth, gene regulation, and processes that rely on proton motive force [42–44]. Biosynthesis of UQ-8 takes place via a highly conserved pathway that involves a large number of genes (i.e., *ubi*) that have been identified by genetic studies [45]. The *ubiCA* mutants all had altered patterns of UQ-8 biosynthesis and were hypersensitive to $H_2O_2$; production of both $O_2$ and $H_2O_2$ was significantly higher in the mutants than in the wild-type strain [46]. In our study, *VP0097* (*ubiB*) and *VP0315* (*ubiX*) mutants showed similar high sensitivities to UVA-LED that did not correlate with changes in UQ-8 concentrations (Figure 6). The *ubiX* gene product catalyzes decarboxylation as part of the biosynthesis of UQ-8; inactivation of the gene leads to diminished levels of UQ-8 in target *E. coli* cells [47]. By contrast, *ubiB* is believed to be required for the first monooxygenase step in UQ-8 biosynthesis, although the detailed roles played by each of the gene products remain unclear [48]. Taken together, the combined results suggest that functions of the *ubi* gene product may be related to the degree of UVA sensitivity.

## 5. Conclusions

In conclusion, genes associated with $F_0F_1$-type ATP synthase, the general secretion pathway, and ubiquinone synthesis were all identified as candidate genes that modulate sensitivity to UVA irradiation. A comprehensive review of the genes and gene products modulating these responses will provide a remarkably improved understanding of the efficiency of sterilization procedures and their application to resistant bacteria. For example, a combination of UVA-LED irradiation and treatment of inhibitors against the identified gene products, such as a bacterial $F_0F_1$-type ATP synthase inhibitor piceatannol, which inhibits growth of *E. coli* and other bacteria [49], may be a good strategy to increase the efficiency of the bactericidal effect. Further analysis of several of the critical features and functions of these target genes will be required.

**Supplementary Materials:** The following are available online at http://www.mdpi.com/2076-3417/10/16/5549/s1, Figure S1: Emission spectrum of various UV lamps, Figure S2: A flow chart of the experiment of L-serine treatment for UVA-sensitivity, Figure S3: Sensitivies of mutants from first-round selection to UVA-LED, and Figure S4: Sensitivities to hydrogen peroxide in highly UVA-sensitive *V. parahaemolyticus* mutants.

**Author Contributions:** Conceptualization, K.M., Y.K., and A.T.; methodology, M.M.-Y., K.M., M.A. (Mutsumi Aihara), T.S., and T.U.; software, T.E., and M.A. (Masatake Akutagawa); validation, M.M.-Y., K.M., and A.T.; formal analysis, M.M.-Y., K.M., A.H., and T.K.N.B.; investigation, M.M.-Y., K.M., and A.H.; resources, E.T., and M.A.; data curation, M.M.-Y., and K.M.; writing—original draft preparation, M.M.-Y., and K.M.; writing—review and editing, K.M., and A.T.; visualization, M.M.-Y., and K.M.; supervision, A.T.; project administration, A.T.; funding acquisition, A.T. All authors have read and agreed to the published version of the manuscript.

**Funding:** This work was supported by JSPS KAKENHI Grant Numbers 20H01616.

**Acknowledgments:** The excellent technical assistance of Hideaki Maseda (National Institute of Advanced Industrial Science and Technology) is gratefully acknowledged. We thank Maki Uwate, Miyuki Edagawa, and Natsumi Iwamoto (Tokushima University) for technical assistance. This study was supported by Support Center for The Special Mission Center for Metabolome Analysis, School of Medical Nutrition, Faculty of Medicine of Tokushima University, and Support Center for Advanced Medical Sciences, Tokushima University Graduate School of Biomedical Sciences. This work was supported by the Research Clusters program of Tokushima University.

**Conflicts of Interest:** The authors declare no conflict of interest.

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
