# Peer review of "Identification of Genes Associated with Sensitivity to Ultraviolet A (UVA) Irradiation by Transposon Mutagenesis of Vibrio parahaemolyticus"

_applsci, doi:10.3390/app10165549_

Round 1

Reviewer 1 Report

The research presented in the manuscript titled "Identification of genes associated with sensitivity to ultraviolet A (UVA) irradiation by transposon mutagenesis of Vibrio parahaemolyticus" aimed to identify genes associated with UVA sensitivity.

General comments

The manuscript is well written and contains a description of a correctly planned experiment. Some detailed comments are provided below. In order to improve the applicability of the research, the justification of the gained knowledge should be added. For example, how the results could be utilized in designing new disinfection methods?

Detailed comments

Line 65: The numbering of references should be checked. It should be rather 14 than 12. Generally, the reference numbering requires revision because the next cited articles do not exactly correspond with the sentence content i.e. [17,18].

Lines 77-94: Please, revise the text and remove all methodology and results in the next sections. I suggest editing the goal and list specific tasks, e.g. mutant selection, ATP determination, etc.

Author Response

Thank you for your valuable comments. We answered the reviewer’s comments as described below.

General comments: The manuscript is well written and contains a description of a correctly planned experiment. Some detailed comments are provided below. In order to improve the applicability of the research, the justification of the gained knowledge should be added. For example, how the results could be utilized in designing new disinfection methods?

A1. We agreed your comment. We inserted our opinion about new disinfection methods by the results of this study, ‘For example, a combination of UVA-LED irradiation and treatment of inhibitors against the identified gene products, such as a bacterial F0F1-type ATP synthase inhibitor piceatannol which inhibits growth of E. coli and the other bacteria [49], may be a good strategy to increase the efficiency of bactericidal effect.’, at Lines 379-382 in the revised manuscript.

Detailed comment #1: Line 65: The numbering of references should be checked. It should be rather 14 than 12. Generally, the reference numbering requires revision because the next cited articles do not exactly correspond with the sentence content i.e. [17,18].

A2. We were sorry our incorrect citation. We changed the reference #12 and #17 into #14 and #18, respectively, in the revised manuscript.

Detailed comment #2: Lines 77-94: Please, revise the text and remove all methodology and results in the next sections. I suggest editing the goal and list specific tasks, e.g. mutant selection, ATP determination, etc.

A3. We agreed your comments. We moved the sentences into the Results and the Discussion sections. And we added the goal of this study at the end of the Introduction section in the revised manuscript.

Reviewer 2 Report

The purpose of the study was to identify genes associated with UVA sensitivity with the goal of improving UVA‐LED mediated bactericidal activity for Vibrio parahaemolyticus mutants.

It is a scientifically sound paper and of high interest for scientists working in food and water areas.

The study is of High interest also for industrials for a future potential application.

Molecular techniques were used for mutants selection .

The authors  developed a UVA irradiation disinfection system based on a light‐emitting diode
(UVA‐LED) .THe UVA irradiation method induces cellular membrane damage and indirectly results in growth delay by increasing intracellular levels of reactive oxygen species (ROS), including superoxide anion radicals , hydroxyl radicals,hydrogen peroxide , and singlet oxygen.All different factors were studied by appropriate methodology.

Results are well presented and discussion is done in depth.

Material and Methods are well described and reproducible and bibliography is up to date.

Delete the word growth (in double) from line 68.

My suggestion is to ACCEPT the paper in its present form.

Author Response

The authors developed a UVA irradiation disinfection system based on a lightemitting diode (UVA-LED) . The UVA irradiation method induces cellular membrane damage and indirectly results in growth delay by increasing intracellular levels of reactive oxygen species (ROS), including superoxide anion radicals, hydroxyl radicals, hydrogen peroxide, and singlet oxygen. All different factors were studied by appropriate methodology.

Results are well presented and discussion is done in depth.

Material and Methods are well described and reproducible and bibliography is up to date.

Delete the word growth (in double) from line 68.

Thank you for your valuable comments. We deleted the word ‘growth’ from Line 67 in the revised manuscript.

Reviewer 3 Report

The article entitled “identification of genes associated with sensitivity to ultraviolet A (UVA) irradiation by transposon mutagenesis of Vibrio parahaemolyticus” is an interesting article well written and presented, that sheds light on the genes involved in the UVA irradiation resistance. The paper could be accepted after minor revisions.

Introduction

Lines 46-51 and lines 52-63. This paragraph for my opinion should be rewritten. The link between these decontamination methods (Chlorination and ozonation) and those mentioned later in the text (UV treatments) is unclear.

Line 65. Please check and revise this reference, it would seem incorrectly cited. Also check the reference 17.

Lines 80-94. The authors could shorten this paragraph and many sentences should be moved on the discussion section.

Material and Methods

Line 107-108. Please check and revise, also in relation with that reported in line 165. A flow chart could better clarify the experimental procedure.

Results and discussion

In figure 4-a a higher variability of ATP concentration in WT cells was found respect to the standard deviation of mutants. How do the authors explain this fact?

Author Response

Thank you for your valuable comments. We answered the reviewer’s comments as described below.

Q1. Introduction: Lines 46-51 and lines 52-63. This paragraph for my opinion should be rewritten. The link between these decontamination methods (Chlorination and ozonation) and those mentioned later in the text (UV treatments) is unclear.

A1. We agreed you comment. We modified the paragraph about the decontamination methods into ‘Chlorination and ozonation have high efficiency against bacteria in general. However, some health problems have been observed. For instance, residual chlorine in drinking water can cause the formation of potentially carcinogenic halogenated by-products [10]. Likewise, ozonation can lead to the formation of high concentrations of undesired by-products, including bromates, which are also potential human carcinogens [11]. Other well-known disinfection methods include sunlight and ultraviolet (UV) irradiation; these modalities produce by-products but these reported so far are below the level of health concerns [12]’ at Lines 46-52, Page 2, in the revised manuscript.

Q2. Introduction: Line 65. Please check and revise this reference, it would seem incorrectly cited. Also check the reference 17.

A2. We were sorry our incorrect citation. We changed the reference #17 into #18 in the revised manuscript.

Q3. Introduction: Lines 80-94. The authors could shorten this paragraph and many sentences should be moved on the discussion section.

A3. We agreed your comment. We stayed only one sentence in the paragraph and moved the other sentences into the Results and the Discussion sections.

Q4. Material and Methods: Line 107-108. Please check and revise, also in relation with that reported in line 165. A flow chart could better clarify the experimental procedure.

A4. We agreed your comment. We moved the sentence at Lines 107-108 in original manuscript into the Section 2.4. in the revised manuscript. And we added a flow chart about the experiment of L-serine treatment in supplementary Figure S2 of the revised manuscript.

Q5. Results and discussion: In figure 4-a a higher variability of ATP concentration in WT cells was found respect to the standard deviation of mutants. How do the authors explain this fact?

A5. We understood your comment. We run seven independent experiments (n = 7) of ATP concentration in WT in Figure 4a. A concentration, 1315.5 umol/mg protein, was furthest one, but not a significant outlier (P > 0.05) by Grubbs' test. So ATP concentration in WT cells had high variability in the figure.
